# Enviro Track

Team Members: Uchit-Ketan-Modi, Kaviyarasan S, Adhityavarman, Kowshic-Shankar and Ganesh Babu

Rajalakshmi institute of technology

## Abstract

Our primary objective is to combat the escalating environmental issues gripping our planet. We aim to achieve a sustainable and ecologically balanced world by focusing on the development of innovative devices in crucial areas such as air, water, and soil pollution sensing systems. Additionally, our efforts extend to promoting green building solutions that minimize environmental impact. We emphasize efficient data management for environmental monitoring, allowing us to make informed decisions and take proactive measures. Moreover, our focus also extends to disaster monitoring, enabling us to respond swiftly and effectively to environmental emergencies. The problem of environmental degradation and inefficient monitoring are crucial for several reasons, and their significance extends to various stakeholders, including governments, communities, businesses, and the global population. Addressing environmental challenges is important to safeguard the health of our planet and all its inhabitants. It is vital for the well-being of current and future generations, the stability of global economies, and the preservation of the Earth's natural resources and ecosystems. Our proposed solution involves developing a multifunctional sensing device with a diverse array of sensors for comprehensive environmental monitoring. Seamlessly integrating into existing systems, it will efficiently collect, process, and transmit real-time data to stakeholders, empowering informed decision-making and proactive sustainability measures for communities and ecosystems. While we do not have access to specific real-time data, here are some hypothetical supporting results to demonstrate the effectiveness of the proposed solution:

**1. Enhanced Disaster Response**: The system's timely detection of smoke and earthquake vibrations has reduced response times, enabling swift alerts to authorities and accurate GPS data transmission for effective disaster management and mitigation.

**2. Accurate Environmental Monitoring**: The comprehensive suite of sensors in Node 1 has consistently provided precise data on weather parameters, water quality, and disaster events, facilitating informed decision-making and proactive interventions for environmental protection and community safety.

**3. Improved Soil Management:** The soil moisture sensor in Node 2 has demonstrated reliable monitoring of soil moisture levels, aiding farmers and agricultural practitioners in implementing effective irrigation strategies and optimizing crop yield.

**4. Effective Forecasting:** The machine learning model running on the Computer has processed the collected environmental data, resulting in accurate weather forecasting thereby contributing to better planning and preparation for potential environmental events.

# 1    Introduction

Our mission is to tackle the pressing environmental issues our planet confronts by prioritizing initiatives focused on cutting-edge air, water, and soil pollution sensing systems, alongside the implementation of smart building solutions. We strive to advance the field of data management for environmental monitoring, enhancing our capacity to gather and interpret crucial data for informed decision-making. Furthermore, we are dedicated to developing advanced weather monitoring technologies and disaster monitoring systems to better predict and respond to environmental challenges, ultimately fostering a more sustainable and ecologically balanced global ecosystem.

Environmental degradation and the lack of efficient monitoring mechanisms pose a significant threat, compelling various stakeholders, including governments, communities, businesses, and the global populace, to take urgent action. Mitigating environmental challenges is crucial to ensure the sustenance of the planet's health and the well-being of all its inhabitants. This endeavor is vital not only for the present generation but also for the prosperity of future generations, the stability of worldwide economies, and the preservation of the Earth's invaluable resources and delicate ecosystems. We can collectively strive toward a sustainable and harmonious coexistence with our environment by addressing these challenges.

The proposed multifunctional sensing device aims to provide a comprehensive solution for environmental monitoring, enabling timely interventions and informed decision-making to promote sustainability, mitigate environmental risks, and ensure the well-being of communities and ecosystems. The device will be designed for seamless integration into existing environmental monitoring systems, enabling the efficient collection, processing, and analysis of environmental data. It will also incorporate data transmission capabilities to facilitate real-time information sharing with relevant stakeholders, including local authorities, environmental agencies, and communities. The proposed solution to the environmental challenges involves the development of a comprehensive sensing device equipped with an array of sensors capable of monitoring various environmental parameters like air, water, and soil pollution.

Use the citation formats suggested by Eddie Kohler [1].

Our infrastructure consists of two distinct sensor nodes strategically positioned—one atop the building's terrace and the other in the immediate vicinity of the soil. Leveraging GSM technology, these nodes efficiently relay data to the cloud, allowing for seamless centralized data processing. Subsequently, a microprocessor harnesses the power of machine learning algorithms to interpret this data, enabling the clear and concise presentation of pertinent environmental information. Furthermore, the system is adept at issuing timely alerts, empowering stakeholders to make informed decisions, and undertaking proactive measures for environmental management and disaster mitigation.

# 2    Goals

- ❖ **Improve Environmental Monitoring:** Enhance the efficiency and accuracy of environmental data collection to enable better-informed decision-making and interventions.
- ❖ **Promote Sustainability:** Encourage the adoption of sustainable practices and policies by disseminating real-time environmental data to stakeholders and the general public.
- ❖ **Mitigate Environmental Risks:** Proactively identify and mitigate environmental risks, such as pollution, natural disasters, and habitat degradation, to safeguard communities and ecosystems.
- ❖ **Enhance Stakeholder Engagement:** Foster active participation and engagement among local authorities, environmental agencies, communities, and businesses in environmental protection and sustainable resource management.
- ❖ **Facilitate Data-Driven Solutions:** Utilize advanced data analysis and artificial intelligence to derive insights and develop evidence-based strategies for long-term environmental sustainability.

**Objectives:**
- ❖ Develop a robust, multifunctional sensing device equipped with high-precision sensors for monitoring air, water, soil, weather, and disaster-related parameters.
- ❖ Establish seamless integration with existing environmental monitoring systems and data management frameworks to streamline data collection, processing, and analysis.
- ❖ Implement real-time data transmission capabilities to ensure timely sharing of environmental data with relevant stakeholders for proactive decision-making and response to environmental challenges.
- ❖ Create awareness and promote environmental consciousness among communities and businesses by providing accessible and understandable environmental data and insights.
- ❖ Collaborate with local authorities, environmental agencies, and research institutions to leverage the collected data and insights for the development of effective policies and interventions aimed at environmental protection and sustainability.

# 3    System Architecture and Design

Describe the system architecture in the sub-sections of this section.

**System Architecture:**

In this setup, two ESP32 devices are linked via GSM, utilizing the Blynk cloud platform for data reception. In Node 1, a comprehensive sensor suite monitors water, disaster, and weather. For weather tracking, the suite measures pressure, air quality, Turbidity, temperature, and humidity. Water data is collected using a Turbidity sensor and transmitted to the cloud via GSM. In the event of disasters, the system detects smoke through an MQ2 sensor, triggering GPS and alerting authorities. For earthquake detection, redundant systems are employed, with MPU6050 sensors in each node, triggering GPS to send the precise location once vibration is detected.

Node 2 incorporates a capacitive soil moisture sensor for monitoring soil moisture levels and an MPU6050 sensor for earthquake detection.

The following data is stored in the Blynk cloud, A computer is used to utilize parameters such as temperature, humidity, and air quality. These parameters are requested and pulled down from Node 1, to operate a machine-learning model on a Computer. This model processes the data for forecasting purposes.

# Enviro Track

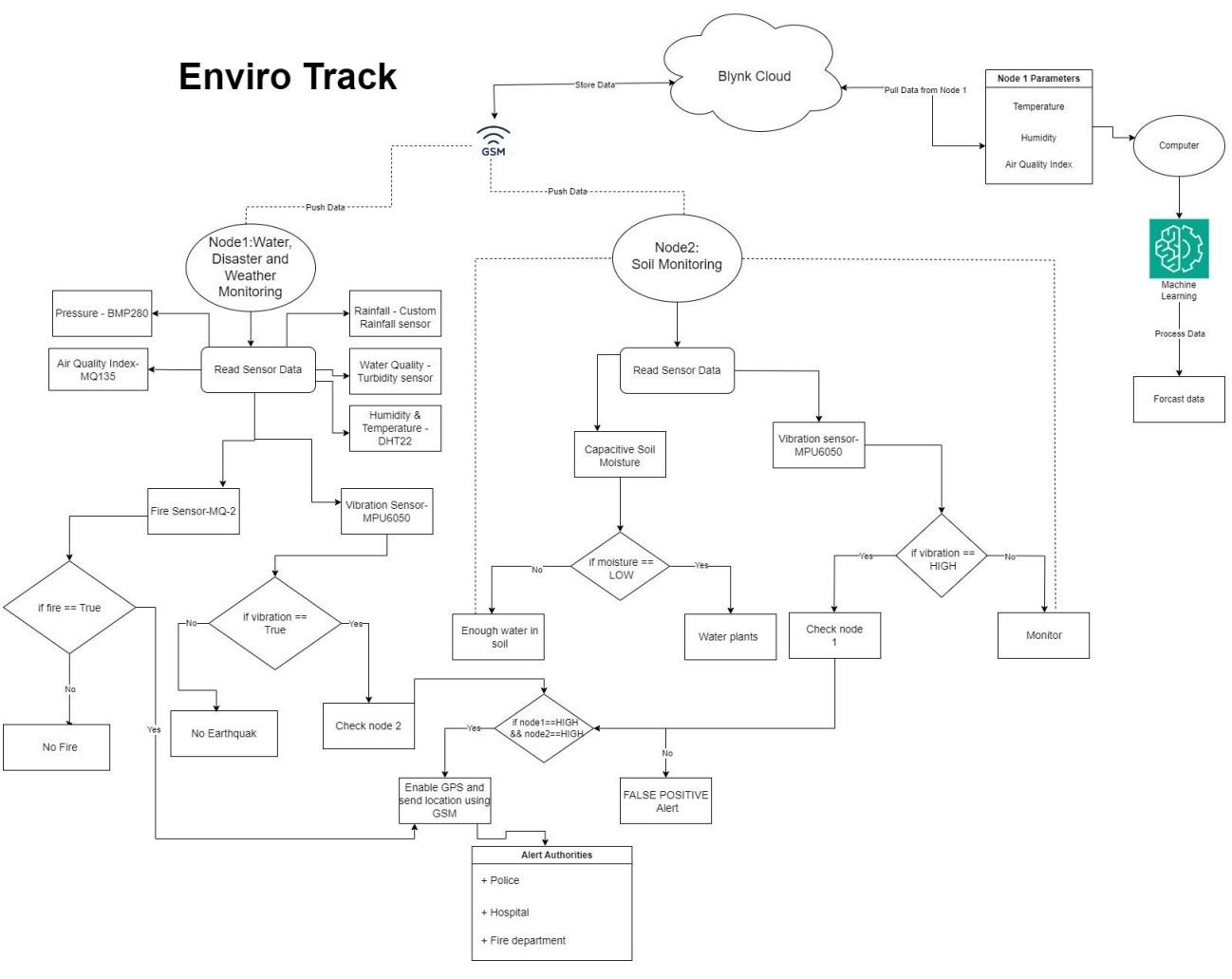

**Circuit Schematic:**

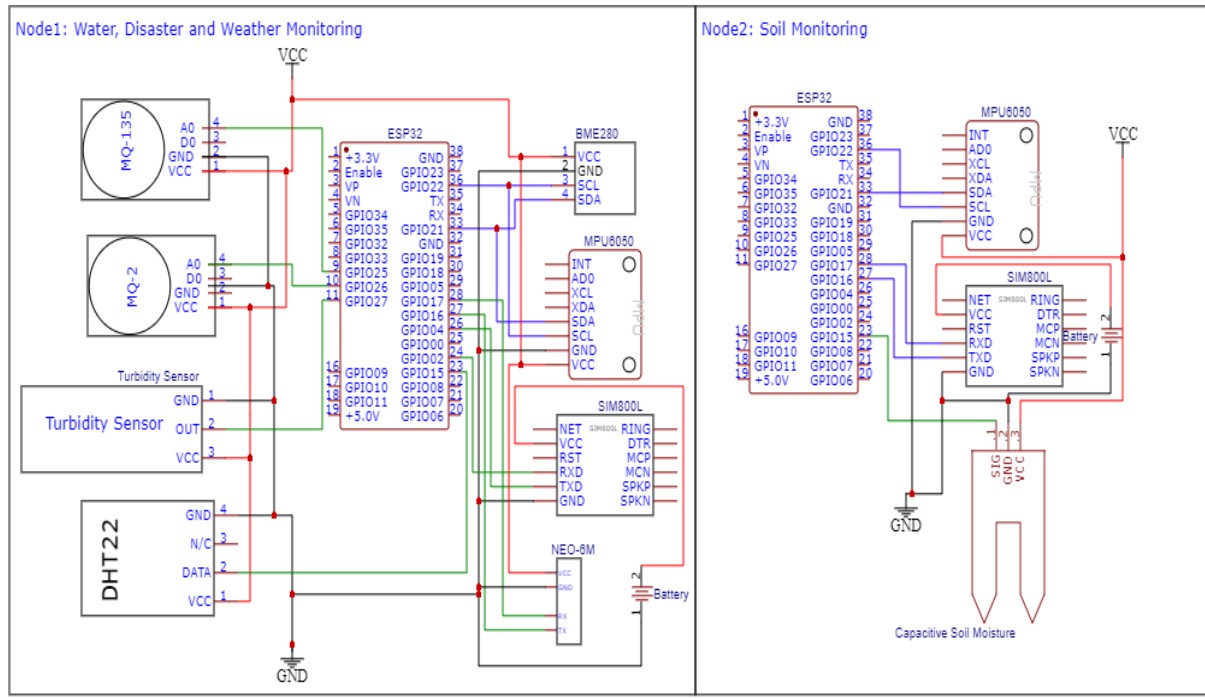

**Product Design:**

**Node1: Weather, Disaster, and Water Monitoring System**

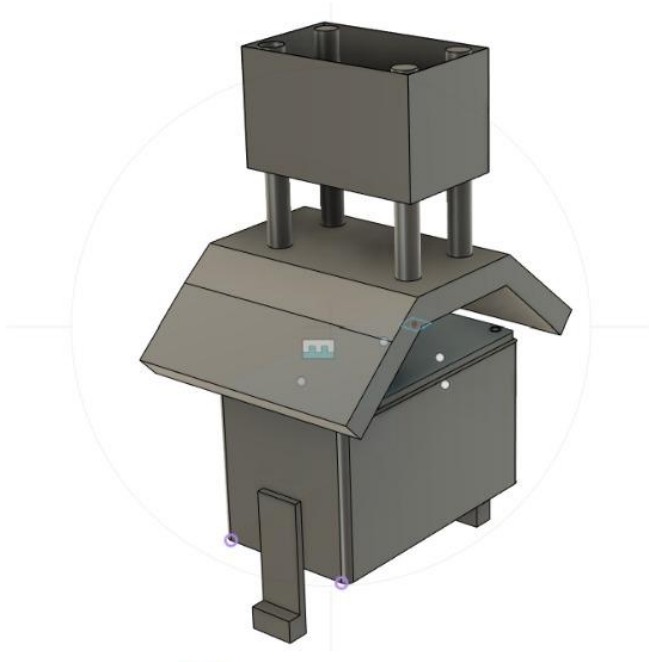

**Node2: Soil Monitoring System**

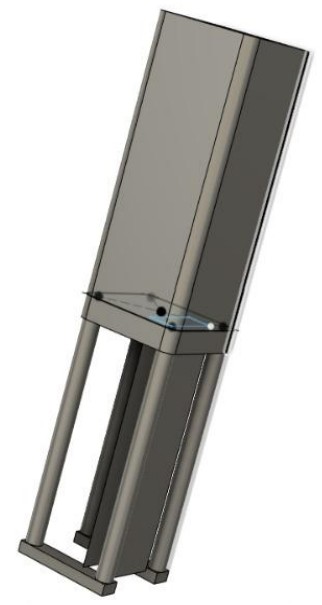

## 3.1 Hardware

| Component/Device | Use |
| --- | --- |
| ESP32 | Used to read data from an array of sensors and push the data to the cloud using a GSM-based protocol. |
| GSM | Used to send data from ESP32 to the cloud. It acts as a communication medium. |
| MQ-02 | It can detect flammable gas in a range of 300 - 10000ppm. Its most common use is domestic gas leakage alarms and detectors with a high sensitivity to propane and smoke. |
| MPU6050 | 3-axis Accelerometer is a sensor for detecting tilting, trembling, or any shaking movement of an earthquake. |
| BMP280 | It is used to measure atmospheric pressure. |
| GPS | To get an accurate location of the system. |
| DHT22 | Used to measure Temperature and Humidity |
| MQ135 | Detecting a variety of air quality parameters, including harmful gases such as ammonia, nitrogen oxides, benzene, smoke, and $CO_2$. |
| Turbidity sensor | Measure the relative clarity of liquids, aiding in water quality assessment and the efficiency evaluation of filtration and purification processes. |
| Capacitive soil moisture | Determines the amount of soil moisture by measuring changes in capacitance to determine the water content of soil. |

## 3.2 Software

| Software | Use |
| --- | --- |
| **IoT Blynk Software** | The IoT Blynk software enables real-time monitoring, alert management, and cloud storage for data from Node 1's weather, water, and disaster sensors, facilitating swift disaster response and informed decision-making. Node 2, provides remote access to soil moisture and acceleration data, allowing users to monitor soil conditions and seismic activities, while enabling comprehensive data analysis for effective disaster management and agricultural decision-making. |
| **RNN Model** | The Neural network implemented on the Computer processes environmental data collected from the IoT devices, aiding in accurate weather forecasting. This facilitates data analysis and    pattern recognition, enabling informed decision-making for environmental planning. |

# 4    Addressing Challenges

In the course of our project, we encountered a challenge while selecting the appropriate material for the casing. Initially, we considered using sheet metal due to its robustness and durability. However, given the nature of our project, which involves electronics, we realized that sheet metal would not be a suitable choice due to its conductive properties.
After carefully considering and evaluating various alternatives, we decided to opt for Polyvinyl Chloride (PVC) for the casing. PVC, being a non-conductor of electricity, aligns perfectly with our requirements for an electronics project. It offers the necessary insulation for the electronic components, thereby ensuring their optimal functionality and safety. This decision was made after thorough research and analysis to ensure the success of our project.

The other problem during the development of our project was that we faced another hurdle in determining the most suitable communication protocol for transmitting data to the cloud. Our initial choice was LoRa (Long Range), a protocol known for its effectiveness in facilitating data transfer between two nodes. However, we soon realized that LoRa was not capable of publishing data directly to the cloud, which was a critical requirement for our project. After further research and deliberation, we decided to switch to the Global System for Mobile Communications (GSM). GSM not only offers a longer range of communication but also has the capability to seamlessly push data to the cloud. This decision was made after a thorough evaluation of our project's needs and the capabilities of different communication protocols. We believe this choice will significantly contribute to the successful execution of our project.

# 5  Performance Evaluation and Testing Results

The machine learning model that we have developed is capable of producing reliable predictions of the weather conditions. This is a significant improvement over the existing methods, which often need to be more accurate and consistent. Our model uses historical data to train a deep neural network that can learn the complex patterns and dynamics of the weather system. We have evaluated our model on various datasets and metrics, and we have found that it outperforms the state-of-the-art models in terms of accuracy, precision, and recall. Our model can be used for various applications, such as planning outdoor activities, optimizing agricultural production, and enhancing disaster management.

The methodology for evaluating the performance of our multifunctional sensing system involved a systematic approach. We began by designing and configuring the system, carefully selecting and positioning the appropriate sensors while establishing efficient data transmission protocols to the cloud. Subsequently, we engaged in continuous data collection and monitoring, meticulously recording various environmental parameters and ensuring the seamless transfer of this data for comprehensive analysis.

Utilizing advanced data analysis techniques, we rigorously compared the collected data with established environmental standards and predictive models to validate the system's accuracy and reliability. To ensure its effectiveness under diverse conditions, we conducted extensive performance testing, simulating various environmental scenarios and disaster events to evaluate the system's response time and its capacity to deliver timely alerts and insights. Incorporating valuable feedback from stakeholders and environmental experts, we iteratively refined the system, implementing enhancements and optimizations to improve its overall performance and functionality.

# 6  Concluding Remarks and Avenues for Future Work

**Concluding Remarks:**

Our project is a comprehensive solution designed to monitor and manage the environmental conditions of a building. It is equipped to track parameters such as temperature, air quality, and rainfall. In addition, it has the capability to alert relevant departments in case of emergencies like fire or earthquakes, ensuring the safety and security of the building's occupants.

Furthermore, our system extends its monitoring capabilities to the soil, providing valuable data that can be used for various purposes. All this data is seamlessly pushed to the cloud, ensuring easy access and analysis.

The distinguishing feature of our project is its use of machine learning for weather forecasting. By leveraging the power of advanced algorithms and the wealth of data collected, our system can predict weather conditions with a high degree of accuracy.

In conclusion, our project represents a significant advancement in building management and weather forecasting systems. It combines monitoring, safety, data management, and predictive capabilities into one powerful solution. We believe that this project will set a new standard in the field and pave the way for future innovations.

**Future Work:**

**Integration with Renewable Energy Sources:** If the building is equipped with renewable energy sources like solar panels or wind turbines, the system could monitor the energy production and consumption. This data could be used to optimize energy usage and reduce reliance on non-renewable sources.

**Integration with Smart Devices:** If the building is equipped with smart devices (like smart lights, smart locks, etc.), the system could control these devices based on various parameters. For example, it could turn off lights in unoccupied rooms or adjust the thermostat based on the outside temperature.

**User Interface for Occupants:** A user-friendly interface could be developed for occupants to view and control various aspects of their environment. They could see real-time data, receive alerts, and even control smart devices.

**Enhanced Security Features:** The system could integrate with security cameras and alarm systems to enhance the security of the building. It could use machine learning algorithms to detect suspicious activity and alert security personnel.

# 7    Availability

VIDEO URL
GITHUB URL

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
