# OpenReview forum: "ENVIRO TRACK"
_helsinki.fi/ESPC/2023/Competition — ESPC 2023 ShortPresentation_

### Official Review · Reviewer_K8u8 · 2023-11-14

**Rating:** 2
**Confidence:** 3

**Summary:**

The authors present their solution that consists of a platform with multiple sensors.
Their system can upload the values monitored by their sensors to a remote server for further analysis, and the results are displayed on a mobile app.

**Strengths:**

- The team is an interdisciplinary team that has looked at the problem from various angles.
- The solution is timely, and the authors have a working prototype that is demonstrated in the video.
- The authors have discussed the challenges in building the system and the factors affecting the design choices such as choosing GSM over LoRa.

**Weaknesses:**

- The report is not polished, and needs proof-reading. There are a couple of typos and a simple spell check can address most of them.
- The abstract is very long and can be merged with the introduction section of the report.
- The authors present a black box approach without discussing the details of the RNN and how it was trained, and there is limited discussion on the usage.
- There is limited discussion on the results and the insights gained by deploying their solution. What were the challenges faced when you deployed the solution, and how do you plan to takle them in the future? Also, who were the experts you contacted, and what were their insights?
- There is limited discussion on the critical analysis of the values reported by the sensors. For instance, what is the ground truth against which the accuracy of the values reported by the sensors is calculated? What is the strategy to account for false positives. For instance, if the node 2 is moved by a human then how do you detect that it is not an earthquake?

---

### Official Review · Reviewer_bJiz · 2023-11-17

**Rating:** 2
**Confidence:** 4

**Summary:**

The project monitors weather, earthquake, and soil moisture with parameters such as pressure, air quality, turbidity, temperature, humidity, and 3-axes acceleration. The project uses ESP32 as microcontroller, GSM for communication and Blynk as cloud platform. Two nodes are created one is placed on roof-top and another one on ground. Proof of concept demo is given in the video, which is decent. It is claimed that RNN based forecasting weather data is done on the data collected.

**Strengths:**

1. Proof of concept demo is good.
2. Good presentation.

**Weaknesses:**

1. No actual deployment and data collection.
2. Sensors have not been calibrated.
3. The sensors deployed are not precision sensors as claimed.
4. No results are shown.
5. No details on RNN are given.

---

### Official Review · Reviewer_pLaM · 2023-11-18

**Rating:** 2
**Confidence:** 4

**Summary:**

The project develops a multifunctional sensing device with a diverse array of sensors for comprehensive environmental monitoring. The project aims to develop sensing systems for soil moisture, earthquake, and other variables. The project uses GSM (unknown network generation either 3Gn 4G or 5G) for data transmission and then pushes the data to Blynk cloud. The video also shows a mobile app (which is not referred to/explained in the report) that is used to show the sensor readings in real-time.

**Strengths:**

The report presents an IoT system to monitor the environment. Good starting work for bachelor students. The high motivation behind the project. The presentation in the video is more informative than the report.

**Weaknesses:**

The project is not novel (The novelty of the project has not been addressed in the report). The project uses off-the-shelf ESP32 devices to establish a simple IoT system.
The report does not say if they have used 4G systems. They say GSM. Overall, many technical details are missing.
The report says they transmit data to Blynk cloud, but in Abstract they say they do not have access to the real-time data. Also, they do not show any measured data.
The report is not written well. There are errors in the report, for example, what is this?:
Use the citation formats suggested by Eddie Kohler [1].

---

### Official Review · Reviewer_bKNs · 2023-11-18

**Rating:** 1
**Confidence:** 4

**Summary:**

In this work, authors have proposed to develop a multifunctional sensing device that is capable of monitoring various environmental issues.
Their aim is to demonstrate applicability of the prototype for disaster response, monitoring weather, water, soil, and effective forecasting. They have proposed to develop two node prototype - one on the top of a building and the other one in the soil.
They have used following sensors - GPS, atmospheric pressure sensor, temperature and humidity sensor, air quality sensor that senses the gases such as, ammonia, nitrogen oxides, benzene, smoke, and CO2, turbidity sesnor, and soil moisture sensor.
Collected data is send to cloud for further processing via IoT Blynk Software.
They also propose a neural network for weather forecasting.

**Strengths:**

One stop sensing device for monitoring various environmental issues.

**Weaknesses:**

Typing mistakes in the report. For example on Page 2 - "Use the citation formats suggested by Eddie Kohler [1]."

After reading the report it appears that the prototype is still in "proposed" state. It is not yet implemented and that is why authors mention on Page 1 - "While we do not have access to specific real-time data, here are some hypothetical supporting results to demonstrate the effectiveness of the proposed solution:"
Without a real-prototype, it is hard to evaluate the effectiveness of the proposed solution, especially when it is aiming to solve a multitude of environmental challenges at once.

As of now, the work severely lacks the novelty.